# Comprehensive Characterisation of a Low-Frequency-Vibration Energy Harvester

**DOI:** 10.3390/s24123813

**Published:** 2024-06-13

**Authors:** Aitor Plaza, Xabier Iriarte, Carlos Castellano-Aldave, Alfonso Carlosena

**Affiliations:** 1Engineering Department, Public University of Navarre (UPNA), Campus of Arrosadia, 31006 Pamplona, Spain; xabier.iriarte@unavarra.es; 2Institute of Smart Cities (ISC), Public University of Navarre, Arrosadía Campus, 31006 Pamplona, Spain; alfonso.carlosena@unavarra.es; 3Department of Electrical, Electronic and Communications Engineering, Public University of Navarre, Arrosadía Campus, 31006 Pamplona, Spain; carlos.castellano@unavarra.es

**Keywords:** energy harvesting, vibration harvester, model identification, low-frequency vibrations, wind turbines

## Abstract

In this paper, we describe a measurement procedure to fully characterise a novel vibration energy harvester operating in the ultra-low-frequency range. The procedure, which is more thorough than those usually found in the literature, comprises three main stages: modelling, experimental characterisation and parameter identification. Modelling is accomplished in two alternative ways, a physical model (white box) and a mixed one (black box), which model the magnetic interaction via Fourier series. The experimental measurements include not only the input (acceleration)–output (energy) response but also the (internal) dynamic behaviour of the system, making use of a synchronised image processing and signal acquisition system. The identification procedure, based on maximum likelihood, estimates all the relevant parameters to characterise the system to simulate its behaviour and helps to optimise its performance. While the method is custom-designed for a particular harvester, the comprehensive approach and most of its procedures can be applied to similar harvesters.

## 1. Introduction

Energy harvesting from ambient sources has been a topic of intense research for at least the last twenty years. With the advent of the Internet of Things paradigm, the need to power up an exponentially increasing number of autonomous sensing nodes has driven intense research to exploit all the available sources of energy in the most diverse environmental contexts [1,2]. The plethora of harvesters described in the literature, in both academic papers and patents, are difficult to classify and categorise. The source of energy, whether it is the Sun (light), vibration, electromagnetic (EM) signals, heat, or sea waves, is just a discriminating first approach to the problem, but it does not help much to dig into the variety of physical principles, materials, devices, mechanical arrangements and electrical and electromagnetic circuits that combine to solve a particular application. The proof of that is that there are a limited number of books or tutorial papers, which mostly have a compilation character, but the bulk of the information is contained in research papers scattered in journals of different character [3,4,5,6].

In this paper, we will restrict our focus to vibration-based harvesters [7]. More precisely, we are interested in those operating in the ultra-low-frequency range [8]. This range is not precisely defined in the literature, but most authors consider the ultra-low-frequency range as that below 10 Hz. There are also a few devices that operate in the sub-hertz range, such as the one we will showcase here. The typical applications for these devices are energy scavenging from human movement, sea waves, high-rise buildings and wind towers. In our case, we are interested in harvesters operating in wind turbines. Such structures need to be continuously monitored to asses their structural health in order to detect or predict catastrophic failures and to estimate their remaining operating life-time. This must be accomplished by a network of acceleration sensors, which provide data later processed by means of Operational Modal Analysis techniques [9]. In most cases, the sensors must be autonomous in energy terms, but even reasonable-sized batteries do not provide enough energy for the time-span needed. Therefore, harvesting devices are mandatory. Vibration, or rotation in some cases, is the only source of energy available, but the characteristic frequencies are below those found in similar applications such as human movement or sea waves [10]. The need to capture multidirectional vibrations is another specific characteristic not present in the mentioned applications. Therefore, the application at hand poses serious challenges to the design of dedicated harvesters. Apart from solutions translated from other problems, which are not optimal, there are a few examples of harvesters specifically designed for wind towers. We can mention Ref. [11], with a harvester for sensors placed inside the wind tower blades, which is an adaptation of a vibration device to rotational operation. More recently, in [12], and in other papers from the same authors, the application to wind turbines is also considered in the proposed vibration harvesters. In our recently published paper, we describe a harvester for wind turbines, which will be the device used here to exemplify the characterisation method we propose. As demonstrated in [13], such harvesters are very competitive. However, in order to improve their efficiency, it would be very interesting to precisely characterise them, which is undertaken in the present paper.

As with most harvesters operating in the ultra-low-frequency range, ours is based on EM conversion [14]. Otherwise, they require mechanisms of frequency up-conversion to make use of piezoelectricity [15]. Therefore, the device is implemented in the form of a moving mass relative to a casing locked to the moving source of energy. Magnets and coils are placed on the moving mass(es) and the casing, respectively, or the other way around, in such a way that the EM interaction in the form of damping is the mechanism of energy (voltage) generation. The physical principles dictate also that the dimensions of such devices be of the order of the displacements induced on them by the moving (vibrating) source of energy. One of the problems when one deals with harvesters is that it is extremely difficult to make comparisons in terms of efficiency, even when restricting to those using similar principles and oriented to very close applications [14]. In general, only the resulting power generation values are provided for particular working conditions. Thus, it is difficult if not impossible to determine to what extent the harvester architecture (mechanical configuration, materials, etc.), the parasitic losses, the power conversion circuitry and the load influence the global efficiency. Moreover, sound and well-accepted figures of merit that take into account scaling factors do not exist [3,8,16]. In the case of ultra-low-frequency-vibration harvesters, it is evident that frequency, acceleration levels, size and weight play a relevant role to set the efficiency levels, but it is not so evident how they interplay.

Therefore, to achieve a complete characterisation of a harvester that enables having knowledge of its practical limits, the sources of energy dissipation, an optimisation of its behaviour and, therefore, a fair comparison with similar harvesters, the following steps must be taken:First, obtain a sensible model of the harvester. If possible, it should consider not only the global behaviour in terms of the energy yield but also model the mechanical/EM/electrical interactions, which would enable detecting limitations and demonstrate means of improvement.A complete experimental characterisation of the device, oriented to validate and characterise the model.An identification procedure to validate the model and extract the relevant parameter values. With these parameters, the model will serve to accurately simulate the harvester behaviour and predict the energy generation capability under different levels of excitation.

A literature search shows that those tasks are rarely accomplished and only partially when they are. Therefore, we describe in this paper a comprehensive characterisation of an ultra-low-frequency EM harvester. There are, for instance, references that concentrate mainly on the experimental setup to characterise the efficiency in terms of the power generated for a given excitation [17,18], irrespective of the harvester design. In some cases, the method also includes the power conversion circuitry, which may reduce the overall efficiency of the harvester [19]. In [20], a piezoelectric harvester is described in terms of the voltage output versus the energy input and modelled with a simplified linear frequency-dependent system. The system is measured and characterised, but no parameters are identified. Only the output impedance is directly measured. However, the experimental measurements are not contrasted in those papers with a model describing the device.

The experimental characterisation of resonant piezoelectric harvesters is the main topic in [21], which describes a complete experimental test bench combining commercial instruments with custom electronic circuitry. The goal is to find the optimum load for the harvester, the method being compatible with the IEC 62830-1:2017 standard [22] for the characterisation of piezoelectric harvesters. The resulting characterisation is non-parametric, providing the frequency response, impedance, etc., of the harvester. An experimental test bench, making use of commercial instrumentation to characterise a particular kind of piezoelectric harvester in terms of the energy generation, is also described in [23]. The work in [24] limits the focus to the particular software development to characterise the piezoelectric materials.

Bearing in mind the general character of the measurement systems described above, it can be understood that they do not assume a particular model for the device to be characterised. In contrast, some low-frequency harvesters based on the EM principle, also denoted as Levitation-Based Vibration Harvesters (LBVHs), propose a model that is, in some way or another, validated by the experimental measurements. This is the case in [25], where a physical white box model is proposed and adjusted to the experiments, but it is not clear how it is created and in what manner the parameters are identified. However, only the global response between the voltage output and input is characterised, and, therefore, the mechanical limits of the device, due to the movement of the proof mass, cannot be established. In [26], a linear LBVH harvester is also modelled, combining physical modelling with numerical modelling, i.e., a grey box model of the magnetic interaction between magnets. The parameters of the EM interaction, which is modelled by a polynomial, are estimated from the simulations of the magnetic field within the harvester. The characterisation of the device is, as in the previous harvesters, in terms of the voltage generated by the harvester versus the excitation, and the process of parameter estimation, if any, is not described. This is also the case for an innovative device that combines the functions of sensing and harvesting [27].

A more elaborated model and the corresponding measurement procedure are described in [28], which works further on a device first proposed originally in [29]. The device is the so-called Preload Snap-Through-Buckling Non-linear Harvester, which comprises a mechanism of frequency up-conversion: a mass suspended on a beam moves back and forth, snapping two piezoelectrics. This movement, actually the stiffness mechanism, is modelled and then measured via optical procedures, and the parameters (coefficients of a polynomial) are identified by the Simplex (Nelder–Mead) algorithm. However, the experimental results do not include a global characterisation of the harvester. Moreover, Ref. [30] also focuses on the modelling of the stiffness of an EM device that results from both physical spring and magnetic repulsion, which is, as in [28], modelled as a high-order polynomial. However, the estimation of the coefficients is based not on measured data but on EM simulations, as in [26].

In this paper, we will characterise one EM harvester whose design and performance were already presented in [13]. First of all, the harvester is modelled by two alternative approaches. They have in common the description of the kinematic movement of the proof masses and their damping due to both friction and the EM interaction between coils and magnets. The elastic force between the magnets, which is strongly non-linear, is described in two alternative ways: by a physically meaningful white box (WB) model based on magnetic dipoles and by a black box (BB) model based on a finite Fourier expansion with no physical meaning. The Fourier expansion is a novel modelling approach, different from the others compiled in [5]. The power generation is also modelled and depends on the damping, position and angular velocity of the masses.

Second, a custom measurement setup is designed and implemented to completely characterise the harvester. On the one side, the displacements of the moving masses under excitation are measured. This may help to establish the upper limits of the mechanical energy available in the harvester. This kind of measurement involves image processing on video frames acquired under controlled excitation conditions. At the same time, the voltages at each of the coils, together with the power generated for different loads, are synchronously measured with a multichannel acquisition system. Once a number of experiments have been carried out, an identification method is applied to obtain the relevant parameters of the model. The identification is mainly based on a maximum likelihood procedure, where the generated power and the position of the masses are measured while the first and second derivatives of the coordinates are calculated by means of regularisation and numerical differentiation. The model is then validated with a set of measurements not used in the identification step.

The paper is organised as follows. In Section 2, we briefly describe the harvester and its modelling. The experimental setup and the identification procedure are described in Section 3. The results of the estimation process are presented and discussed in Section 4, and the conclusions are drawn in Section 5.

## 2. Energy Harvester Description and Mathematical Modelling

### 2.1. System Description

The device at hand is designed to harvest energy from a low-frequency-vibration source. Its working principle is based on proof masses housed within a casing, which is directly connected to the moving structure and vibrates with it. As a detailed description of the harvester can be found in [13], we will limit ourselves to providing the basic information required for a correct understanding of the remainder of the manuscript.

The device, depicted at Figure 1, has a cylindrical housing with a central axis. Attached to this axis by low-friction bearings are three inertial masses with the shape of a wedge (portion of a circle bounded by two radii and an arc).

Each of the individual masses has at its outer end an array of magnets arranged in a Halbach configuration, as shown in Figure 2. This configuration is known for its ability to concentrate magnetic fields in a particular side of the wedge. A beneficial side effect of this arrangement is that the magnetic fields on the sides generate a repulsive force between the adjacent masses. This repulsive effect acts as a non-linear stiffening mechanism that returns the masses to their stable position (evenly spaced at 120°) and reduces the collision between them.

On the other hand, the housing hosts a set of twenty-four coils on its inner curved surface. One half of the housing with its coils is shown in Figure 3. The relative movement of the magnets creates a voltage at the terminals of the coils, which is harnessed to generate power according to Faraday’s law of induction. The design of the device causes each set of magnets to face at least three complete coils, irrespective of their position; see Figure 3.

The housing and chassis of the inertial masses have been fabricated using ABS (Acrylonitrile Butadiene Styrene) plastic with a 3D printer. The Halbach matrix consists of 9 magnets measuring 20 × 5 × 5 mm and 12 magnets measuring 20 × 5 × 2 mm. The coils are composed of 600 turns of 0.1 mm diameter enameled copper wire. Additionally, a printed circuit board that encompasses the fundamental rectification, filtering, and energy storage circuits has been attached to flat sides of the cylindrical structure.

Each one of the proof masses contains a combination of magnets on the outside, and the housing contains a series of coils. Its vibration-induced relative displacement generates an electrical signal in the coils, following the well-known Faraday’s law.

### 2.2. Electromechanical Modelling

Once the physical characteristics of the designed harvester have been presented, we introduce the equations of motion that describe its physical behaviour and that are necessary for its characterisation.

Figure 4 shows a sketch of the device. The centre of the device (point *P*) is defined with respect to the Inertial Frame (IF) (point *O*) by means of two coordinates: x(t) and y(t). The *i*th mass is positioned by an angle $\theta_{i}\left(t\right)$ with respect to the Harvester Frame (HF). For the sake of simplicity, the time dependency of the coordinates from this point onwards has been removed. The centre of gravity (CG) of each of the masses (mass $m_{i}$) is located at a distance $L_{G}$ from point *P*. Based on this, for each moving mass, the position of its CG with respect to the IF is provided by
(1a)$$\begin{array}{cc} x_{i} & =x+L_{G}\cos\left(\theta_{i}\right) \end{array}$$
(1b)$$\begin{array}{cc} y_{i} & =y+L_{G}\sin\left(\theta_{i}\right) \end{array}$$

The system modelling by means of the Euler–Lagrange formulation needs the knowledge of all the energy contributions: kinetic, elastic/magnetic and dissipation. The kinetic energy for each one of the masses is expressed as
(2)$$E_{k_{i}}=\frac{1}{2}m_{i}v_{i}^{2}+\frac{1}{2}I_{zz}{\dot{\theta }}_{i}^{2}$$
where $v_{i}^{2}={\dot{x}}_{i}^{2}+{\dot{y}}_{i}^{2}$, $I_{zz}$ is the moment of inertia with respect to the CG of the *i*th moving mass, and the upper dot represents the time derivative of the variable.

Apart from this, the dissipation of energy arises from two different sources: the interplay between coils and magnets that are connected to an electrical impedance ($R_{i}^{e}$) and the mechanical friction of the moving parts of the device ($R_{i}^{m}$). The former source is the one that serves for energy extraction. Assuming energy dissipation sources are velocity-proportional, the Rayleigh dissipation function can be expressed as
(3)$$R_{i}=R_{i}^{e}+R_{i}^{m}=\frac{1}{2}{\dot{\theta }}_{i}^{2}\sin^{2}\left(12\theta_{i}\right)\;b_{g}+\frac{1}{2}{\dot{\theta }}_{i}^{2}\;b_{0}$$
where $b_{g}$ is the energy generation damping coefficient and $b_{0}$ is the mechanical damping coefficient. While the second term is simply velocity-proportional, the first one is multiplied by a trigonometric factor. The reason for this term to appear is that, in the case of this harvester, the angular separation between magnets is $\frac{\pi }{12}\;rad$, which corresponds to a circular distribution of 12 poles (pairs of magnets) per turn. Therefore, the intensity and direction of the magnetic field generated by each pair along the arc through which it moves (in front of and normal to the coils) approximate a sine of period $\frac{2\pi }{12}\;rad$, and, for each mass–magnet set, the function defining the generated power is proportional to both ${\dot{\theta }}_{i}^{2}$ and $\sin^{2}\left(12\theta_{i}\right)$. Figure 5 shows the relative positions of magnets and coils so that (a) no power is generated and (b) maximum amount of power is generated. For any intermediate position, the power generation would behave as a sinusoidal function of the relative angle $\theta_{i}$.

Finally, the elastic/magnetic energy that comes from the interaction between magnets has been modelled in two different ways.

In the WB model, the magnetic field produced by the magnets acts as if it were a spring with a highly non-linear and position-dependent stiffness. A simplification of the model assumes that magnetic interaction only occurs between adjacent magnets at the extremes of the masses. The magnets, assumed to be dipoles, are located at each lateral face of the masses at a distance *r* from the centre, as shown in Figure 4. If each dipole has a magnetic moment m, the potential energy for a couple of dipoles, also called dipole–dipole energy, is a magnetostatic energy and according to [31] (Equation (A7), Appendix 1) is expressed as
(4)$$E_{p_{12}}=-\frac{{\bar{m}}_{1}{\bar{m}}_{2}}{d_{12}^{3}}\left(\cos\left(\alpha_{1}-\alpha_{2}\right)-3\cos\alpha_{1}\cos\alpha_{2}\right)$$
Then, by adding the three terms (one for each pair) and assuming ${\bar{m}}_{1}={\bar{m}}_{2}={\bar{m}}_{3}=∥m∥$, the total potential energy is simplified to
(5)$$E_{p}=\frac{\nu }{4r^{2}}\left[\frac{8r^{2}-d_{12}^{2}}{d_{12}^{3}}+\frac{8r^{2}-d_{23}^{2}}{d_{23}^{3}}+\frac{8r^{2}-d_{31}^{2}}{d_{31}^{3}}\right]$$
where $\nu ={\mu ∥m∥}^{2}$ takes into account the product of the magnetic moments of the magnets m and the permeability of the medium μ. Moreover, $d_{ij}$ represents the distance between the magnets, as shown in Figure 4.

The BB model for the magnetic interaction is data-driven. Since the magnetic interaction between all the magnets is very complex and fitting a more accurate physical model to represent it is also a complicated challenge [5], the black box model is chosen. That is, a data-driven model where the internal physics of the magnetic interaction are almost unknown. The only evidence for building the model is that the magnetic force basically depends on the separation between the masses ($\theta_{j}$ for j=21,32, and 13). A magnetic interaction model based on polynomials of these variables can be thought as the first natural choice. However, this approach is not practical in this case because, without imposing constraints between the parameters of the polynomials, it would not be possible to obtain the same potential energy for a separation of $\theta_{j}$, $-\theta_{j}$, $2\pi -\theta_{j}$ and $\theta_{j}-2\pi$.

For this reason, the magnetic potential energy has been modelled by a finite Fourier series as follows:(6)$$E_{p}=\sum_{j=21,32,13}\;\sum_{k=1}^{N}a_{j,k}\sin\left(k\theta_{j}\right)+b_{j,k}\cos\left(k\theta_{j}\right)$$
where $a_{j,k}$ and $b_{j,k}$ are some coefficients to be determined and on which the model depends linearly. In our experiments, we have set N=2, which we have found appropriate for this harvester.

The angles $\theta_{21}$, $\theta_{32}$ and $\theta_{13}$ in (Equation 6) are defined as follows:
(7a)$$\begin{array}{cc} \theta_{21} & =\theta_{2}-\theta_{1} \end{array}$$
(7b)$$\begin{array}{cc} \theta_{32} & =\theta_{3}-\theta_{2} \end{array}$$
(7c)$$\begin{array}{cc} \theta_{13} & =\theta_{1}-\theta_{3} \end{array}$$

Working further on the equations of motion, for both models, the Lagrangian for the conservative whole system is
(8)$$L=\sum_{i=1}^{3}E_{k_{i}}-E_{p}$$
and the three equations of motion are obtained as
(9)$$\frac{d}{dt}\left(\frac{\partial L}{\partial {\dot{\theta }}_{i}}\right)-\frac{\partial L}{\partial \theta_{i}}+\frac{\partial R_{i}}{\partial {\dot{\theta }}_{i}}=0,\;\mathrm{for}\;i=1,2,3$$
resulting in three coupled differential equations.

Finally, assuming that the parameters (*m*, $I_{zz}$, *r* and $L_{G}$) of all the masses are equal, the equations of motion of a single mass can be expressed in a compact form as follows:(10)$$\begin{array}{c} \left(mL_{G}^{2}+I_{zz}\right){\ddot{\;\theta }}_{i}+\left(\sin^{2}\left(12\theta_{i}\right)b_{g}+b_{0}\right){\dot{\theta }}_{i}+f_{i}\left(\theta_{i},\theta_{j},\theta_{k}\right)=\tau_{0i} \end{array}$$
where $f_{i}\left(\theta_{i},\theta_{j},\theta_{k}\right)$ is a non-linear function that depends on the angular position of the masses and is different for the WB and BB models. The sub-indices *i*, *j* and *k* vary cyclically, and the excitation term is named $\tau_{0i}$, which is defined as
(11)$$\tau_{0i}=mL_{G}\left(\ddot{x}\sin\theta_{i}-\ddot{y}\cos\theta_{i}\right)$$
where the numeric value of *m* and $L_{G}$ is well-known: the mass *m* is obtained just by weighing it, and position of the centre of gravity $L_{G}$ is obtained by means of CAD (Computer-Aided Design) software.

### 2.3. Power Generation Modelling

It is generally assumed that the mechanism for energy generation in EM harvester is the damping produced by the interaction between the moving (variable) magnetic field and the coils. This is a basic approximate model that assumes ideal coils, which is not the case. There is also dissipative damping due to friction between the moving parts, i.e., bearings, and the common axis. Dissipative damping does not contribute to energy generation and may be relevant in some circumstances. As in the dynamic model, both kinds of damping sources have been modelled separately; the generated power ($\tau_{g}$) can be modelled according to [32] as
(12)$$\tau_{g}=2\sum_{i=1}^{3}R_{i}^{e}=\left(\sum_{i=1}^{3}\sin^{2}\left(12\theta_{i}\right){\dot{\theta }}_{i}^{2}\right)b_{g}$$
where $R_{i}^{e}$ is the electric Rayleigh dissipation function that has already been introduced in (Equation 3) and was used to calculate the Lagrangian of the complete non-conservative system in (Equation 9).

## 3. Model Identification Approach

In the previous section, the different options for accurate modelling of the harvester have been addressed. However, those models are written in terms of unknown parameters that must be estimated in order to characterise the harvester. The parameter estimation process is quite cumbersome and requires several steps to be accomplished.

The very first task to perform is to measure the exciting accelerations, the movement of the masses and the power generated at each coil. To that end, the experimental setup shown in Figure 6 was carried out.

This experimental setup consists mainly of five elements. The device to be identified ➀ is mounted on a specifically designed moving platform ➁ that acts as shaker and is the element that provides the kinetic energy that the device harvests. On the upper part of the platform, mounted on a structure, there is a phone ➂ with a high-speed and high-resolution camera (240 fps). In order to measure the accelerations of the moving platform with respect to the IF, needed in Equation (Equation 11) to obtain the numerical value of $\tau_{0i}$, a triaxial accelerometer (ADXL345) is installed in the electronics of the harvester and is measured together with the power generated by the 24 coils of the harvester. All data are collected using a UART communication protocol through a cabled USB connection and are written to a data table automatically.

The acquisition system is based on a 32-bit Teensy microcontroller at 180 MHz. Its 24 analogue inputs of 12-bit resolution with a full scale of ±1.65 V are connected to the 24 EM generators of the harvester. Simultaneously, the interface board collects data from the triaxial accelerometer via I2C. It also contains sockets to plug-in resistor arrays acting as dummy loads for the generators. Voltages are acquired at a rate of 100 samples per second.

Moreover, a video of the masses moving with respect to the case is recorded while the acquisition system ➃ is running, and, in order to synchronise video and acquisition system, an LED (Light-Emitting Diode) ➄ lights up and is recorded by the camera. Post-processing the video, the acquired signals are synchronised to the video frames using the LED as a trigger. After the video is recorded, it has yet to be processed in order to estimate the positions, velocities and accelerations of the masses.

Once the data have been acquired, the identification process is carried out in four steps, which are detailed below.

### 3.1. Image Processing

In order to identify each of the masses univocally, three different coloured stickers are used, one for each mass, as can be seen in Figure 7.

For every single frame of the video, the areas with specific colours are identified, using built in functions of MATLAB 2023a, and then the centre of gravity of each area is calculated in pixel coordinates.

Basically, this code reads each image from the video and extracts the three main colour channels. Then, for each colour to be identified, it applies a colour threshold and then fills holes in the thresholded image. Finally, by creating a binary image, it segments connected areas, and, for each connected area, its centre is identified. The locations of the masses measured in one of the experiments are shown in Figure 8.

### 3.2. Rotational Coordinates Calculations

As all the masses and stickers rotate with respect to a common axis, the Cartesian coordinates of all coloured areas should fall on a circumference. Fitting the data to the equation of a circumference [33], the radius for the mass rotation and the centre of the harvester are calculated in pixel coordinates. The circle fitted to the measured points of the same experiment is also shown in Figure 8.

Once the fitting is accomplished, the azimuthal positions of each mass are determined using the arc-tangent function. The first row of Figure 9 shows the azimuthal location of the masses for the experiment at hand.

### 3.3. Numerical Differentiation

With the video image processing and accelerations from the sensor, almost all the required information has been collected for the estimation of the model parameters. However, the equations of motion also depend on the first and second derivatives of the azimuthal coordinates. These cannot be measured directly and have to be estimated by differentiation of the calculated azimuthal coordinates. However, this is not a trivial task. In fact, the numerical differentiation of noisy signals still remains as a non-completely solved problem, as shown in [34]. For the smoothing and numerical differentiation of the azimuthal coordinates of the model, the first-order Tikhonov regularisation method [35,36] is used in this paper.

The measured azimuthal *i*th coordinate ($\theta_{m,i}$) for all the experiment instants (ordered by increasing time) can be modelled as their *real* values ($\theta_{i}$) plus measurement noise ($\eta_{i}$) as
(13)$$\theta_{m,i}=\theta_{i}+\eta_{i}$$
As we are looking for a *smooth* representation of the variable, the *variation* of $\theta_{i}$ can be modelled as
(14)$$v_{\theta_{i}}=\frac{1}{\Delta t}\left[\begin{array}{cccccc} -1 & 1 & 0 & \cdots & 0 & 0\\ 0 & -1 & 1 & \cdots & 0 & 0\\ \vdots & \vdots & \vdots & \ddots & \vdots & \vdots \\ 0 & 0 & 0 & \cdots & -1 & 1 \end{array}\right]\theta_{i}=R\theta_{i}$$
(Δt being the time delay between consecutive measurements) in order to minimise the next figure of merit for a given λ:(15)$$V\left(\theta_{i}\right)={∥\theta_{m,i}-\theta_{i}∥}^{2}+\lambda \;{∥R\theta_{i}∥}^{2}$$
The smoothed $\theta_{i}$, using norm L2, is calculated as [35]
(16)$${\hat{\theta }}_{i}={\left(I+\lambda \;R^{\prime }R\right)}^{-1}\theta_{m,i}$$
Then, ${\dot{\theta }}_{i}$ is calculated in terms of the estimated ${\hat{\theta }}_{i}$ as
(17)$${\dot{\theta }}_{i}=R{\hat{\theta }}_{i}$$
Finally, ${\ddot{\theta }}_{i}$ is calculated repeating the procedure for ${\dot{\theta }}_{i}$.

The azimuthal coordinates along with the calculated first and second derivatives of a the previous experiment are shown in Figure 9 for a particular excitation.

### 3.4. Linear Parameter Estimation Approach

Once the acceleration of the platform signal and the image processing have been accomplished, all the necessary data are available for the estimation of the dynamic model parameters, i.e., the inertia parameter ($I_{eq}=m\;L_{G}^{2}+I_{zz}$), the energy generation parameter ($b_{g}$), the friction parameter ($b_{0}$) and the magnetic interaction model parameter (c). It should be noted that although the numerical value of $I_{eq}$ could be obtained a priori, for example by means of CAD software, it may differ slightly from the real value. For this reason, it is introduced as an unknown in the identification process.

Taking the equations of motion previously developed in (Equation 10), it is obvious that they can be written as linear combination of the referred parameters. These equations, which are valid for every time sample, can be written in matrix form as
(18)$$k_{I_{eq}}\;I_{eq}+k_{b_{g}}\;b_{g}+k_{b_{0}}\;b_{0}+k_{c}\;c=\tau_{0}$$
where c depends on the modelling approach, being c=ν for the WB model or $c={\left[a_{j,k}\;,\;b_{j,k}\right]}^{⊤}$ for the BB model.

Gathering the $k_{I_{eq}}$ matrices of the *n* instants of an experiment, and analogously for $k_{b_{g}}$, $k_{b_{0}}$ and $k_{c}$, the following matrices are defined:(19)$$K_{I_{eq}}\;\;=\;\;\left[\begin{array}{c} k_{I_{eq}}^{1}\\ \vdots \\ k_{I_{eq}}^{n} \end{array}\right]\;;K_{b_{g}}\;\;=\;\;\left[\begin{array}{c} k_{b_{g}}^{1}\\ \vdots \\ k_{b_{g}}^{n} \end{array}\right]\;;K_{b_{0}}\;\;=\;\;\left[\begin{array}{c} k_{b_{0}}^{1}\\ \vdots \\ k_{b_{0}}^{n} \end{array}\right]\;;K_{c}\;\;=\;\;\left[\begin{array}{c} k_{c}^{1}\\ \vdots \\ k_{c}^{n} \end{array}\right]\;.$$

Therefore, the matrix equation for *n* time instants is
(20)$$K_{I_{eq}}\;I_{eq}+K_{b_{g}}\;b_{g}+K_{b_{0}}\;b_{0}+K_{c}\;c=\chi_{0}$$
where $\chi_{0}$ gathers the $\tau_{0}$ vectors for *n* instants
(21)$$\chi_{0}=\left\{\begin{array}{c} \tau_{0}^{1}\\ \vdots \\ \tau_{0}^{n} \end{array}\right\}$$

Likewise, the power generation model can be written for *n* instants as
(22)$$K_{b_{g}}\;b_{g}=\chi_{g}$$
where $\chi_{g}$ gathers the $\tau_{g}$ vectors for *n* instants as in (Equation 21).

### 3.5. Comparison to the Previous Identification Procedure

In our previous work [13], the design, fabrication and preliminary identification of this energy harvesting system were carried out. In this work, a series of improvements have been introduced in each step of the estimation process, resulting in a model that represents the real behaviour of the system more accurately. The new features of this work are detailed next.

Two versions of the dynamical model have been realised, a simple one based on magnetic dipole interaction as in [13] and a data-driven one modelled by means of Fourier series. Additionally, a power estimation model has been developed, which is included in the identification process. This power model is dependent on both rotational speed and phase, considering the position of the masses with respect to the individual coils of the generator. In contrast to the previous estimation, the new dynamic models also identify the inertial properties of the masses. Experiments have been performed with different load resistors, and the energy dissipation has been separated in terms of friction (common to all experiments) and generation (specific to each load resistor). The experiments have been performed without restricting the rotation of any of the masses. Unlike [13], where the estimation was performed with a single experiment (transient response), in this work, more than 40 experiments have been performed at different excitation frequencies using some for the estimation process and the rest for the validation process. Finally, the calculation of the numerical derivatives of the mass motion has been carried out by means of the Tikhonov regularisation.

## 4. Experiments and Estimation Results

Once the approach for the model parameter estimation has been defined, the next step requires designing and performing the experiments.

### 4.1. Experiment Design for Parameter Estimation

As has been observed in several experiments, when the system is subjected to harmonic excitation in one direction, one of the masses tends to remain oriented with the direction of excitation. In order to avoid that, all the data with which the estimation is performed have one of the masses stuck around a specific position; the estimation of the parameters is performed using several experiments in which the initial azimuthal positions of the masses are very different. Moreover, these experiments are performed for different excitation frequencies (500, 750, 1000 and 1200 mHz) to ensure that the model is valid for the frequency range for which the device has been designed. Additionally, all these experiments are repeated for several electric resistances *R* (100Ω, 666Ω and 1kΩ), which should result in different values of power generation coefficients ($b_{g}$).

For each combination of electric resistance and excitation frequency, three experiments have been carried out in which the masses are oriented differently in each case. In total, twelve experiments have been carried out for each resistance: eight of them are used for parameter estimation, while the remaining four are used for validation. Furthermore, several additional experiments have been carried out, at different frequencies, without any electrical load. These experiments are devoted to estimating the inherent energy dissipation of the system. Note that, in this case, the resistance value is *∞* and the generation coefficient is $b_{g}=0$.

### 4.2. Combination of Multiple Experiments

Once we have written the sets of Equation (Equation 20) for the electromechanical model and (Equation 22) for the power generation model (for the *n* time instants of a single experiment), it is possible to use all the block matrices for setting the linear system of equations arising from gathering the data from different experiments. Taking the data of the experiments for the 4 load resistors, it is possible to assemble the equations in (Equation 23), where the parameters $b_{g1}$, $b_{g2}$ and $b_{g3}$ related to the different resistors are estimated together and the parameters $I_{eq}$, $b_{0}$ and c are considered common to all the experiments. In (Equation 23), the first row represents an experiment without the load resistors, i.e., infinite resistance; the second to fourth rows represent three experiments, each of them with a different load resistance; the last three rows are written with the power generation model of the same three experiments.

When considering the structure of the matrix in (Equation 23), it is useful to realise that each row of the matrix can represent not only one but the combination of several experiments (see Figure 10). For this purpose, the matrices of the various experiments should be combined analogously to how the submatrices of several time instants are combined to obtain the matrix of a single experiment.
(23)$$\;\;\left[\begin{array}{cccccc} K_{I_{eq}}^{R_{0}} & K_{b_{0}}^{R_{0}} & 0 & 0 & 0 & K_{c}^{R_{0}}\\ -K_{I_{eq}}^{R_{1}} & K_{b_{0}}^{R_{1}} & K_{b_{g}}^{R_{1}} & 0 & 0 & K_{c}^{R_{1}}\\ K_{I_{eq}}^{R_{2}} & K_{b_{0}}^{R_{2}} & 0 & K_{b_{g}}^{R_{2}} & 0 & K_{c}^{R_{2}}\\ K_{I_{eq}}^{R_{3}} & K_{b_{0}}^{R_{3}} & 0 & 0 & K_{b_{g}}^{R_{3}} & K_{c}^{R_{3}}\\ 0 & 0 & K_{b_{g}}^{R_{1}} & 0 & 0 & 0\\ 0 & 0 & 0 & K_{b_{g}}^{R_{2}} & 0 & 0\\ 0 & 0 & 0 & 0 & K_{b_{g}}^{R_{3}} & 0 \end{array}\right]\;\;\left\{\begin{array}{c} I_{eq}\\ b_{0}\\ b_{g1}\\ b_{g2}\\ b_{g3}\\ c \end{array}\right\}\;\;=\;\;\left\{\begin{array}{c} \chi_{0}^{R_{0}}\\ \chi_{0}^{R_{1}}\\ \chi_{0}^{R_{2}}\\ \chi_{0}^{R_{3}}\\ \chi_{g}^{R_{1}}\\ \chi_{g}^{R_{2}}\\ \chi_{g}^{R_{3}} \end{array}\right\}$$

Equation (Equation 23) can be rewritten in a compact form as
(24)Wϕ=χ
In order to estimate ϕ, two circumstances have to be taken into account. First, some components of χ have different dimensions ($\left[\tau_{0}\right]$ = N m and $\left[\tau_{g}\right]$ = N m/s), and, second, variables $\ddot{x}$, $\ddot{y}$ and $\tau_{g}$ are measured with certain errors whose variances are unknown. In this situation, it seems reasonable to determine ϕ as the maximum likelihood estimator using a two-step iterative method (see Section 6.2 of [37]), where the covariance matrix of the measurements is updated for a fixed parameter vector and the parameter estimation is also updated for a fixed covariance matrix. Once the algorithm converges to fixed estimate of parameter vector ϕ and fixed estimate of the covariance matrix of the measurements, it is straightforward to estimate the covariance matrix of the parameters [37].

### 4.3. Estimation Results

As aforementioned, the data obtained from the experiments have been split into two data sets: two thirds of the data are used to estimate the parameters and the rest to validate the obtained results. With the whole set of estimation data, the matrix $W^{est}$ and the vector $\chi^{est}$ are assembled and the vector of estimated parameters $\phi^{est}$ is determined. After that, with the validation data, $W^{val}$ matrix and $\chi^{val}$ are assembled. Finally, to check the accuracy of the identification, a new vector is computed as
(25)$$\chi^{pred}=W^{val}\;\phi^{est}$$
to be compared to the actual measured vector $\chi^{val}$. Both estimation (solving a linear equation) and validation (multiplying matrices) are extremely straightforward and fast to solve.

Figure 10 shows a subset of the vector $\chi^{est}$ for the resistance R=100Ω. One can observe that, for example, between time instants t=60 s and t=80 s, for the same excitation frequency, the masses that are moving are different, as intended.

The accuracy of the estimation can be seen in Figure 11, where, for one of the experiments (R=100Ω and frequency 1.0 Hz), the vector $\chi^{pred}$ obtained with the two models (BB and WB) is compared with the corresponding measured one $\chi^{val}$. As can be seen, the BB model fits the real measurement better. This is due to its larger generality and flexibility because it has more parameters and can take into account other non-linearities that the WB model does not contemplate. Remember that the WB model assumes that the magnetic interaction is produced by dipoles, while this is not exactly the case. The total accuracy of the two methods can be seen in Figure 12, where the histograms of the residual $\chi^{val}-\chi^{pred}$ are shown. For both models, the residual has a mean value that is very small, close to 0.05% of the maximum amplitude of $\chi^{val}$. As can be observed, the BB model has a standard deviation close to 5% of the range of $\chi^{val}$, while, for the WB model, this value doubles.

In parallel to the validation of the dynamic model described in (Equation 10), a validation of the power generation model (Equation 12) has also been performed. For this purpose, Equation (Equation 22) has been evaluated with the data from the different validation experiments with parameters $b_{g1}$, $b_{g2}$ and $b_{g3}$ estimated in Equation (Equation 25), which depend on the resistance selected. For each resistance, the corresponding estimated power $\chi_{g}^{pred}$ has been obtained and compared with the corresponding measured instantaneous power $\chi_{g}^{val}$. Figure 13 and Figure 14 show the power estimation with a resistance of R=100Ω and with two different accelerations with frequencies 1.0 Hz and 0.5 Hz, respectively. In both figures, it can be seen that the two models predict the estimated power identically. This is because Equation (Equation 22) is independent of the magnetic model and therefore so are the estimated $b_{g}$ parameters. For the experiment at 1 Hz, the total energy generated during five seconds is 22.8 mJ if calculated by the estimated model ($\chi_{g}^{pred}$) and 22.4 mJ if calculated based on the power measurements ($\chi_{g}^{val}$). It is worth noting how close the two results are. On the other hand, when the experiment is performed at 0.5 Hz, the measured energy in five seconds is 0.62 mJ and the estimated one is close to 0.73 mJ. In this case, the estimated power is slightly higher than the measured one.

Finally, Table 1 shows the estimated parameters depending on the magnetic model. For the WB model, c equals ν, while, for the BB model, c comprises coefficients $a_{j,k}$ and $b_{j,k}$. Regarding the values of c presented in matrix form, $a_{j,k}$ is located in position (j,2k−1) and $b_{j,k}$ is located in position (j,2k).

### 4.4. Simulation of Generated Power for Different Loads

Once the system has been identified and validated, it is time to evaluate its behaviour under different operating conditions. The main idea is to study, for different accelerations, the dynamic evolution in time of the position of the masses, as well as the power that the system is able to generate.

The first case shows a simulation of the system starting from a position far from its equilibrium position when the external acceleration is zero. In this case, it has been simulated with a load resistance of R=666Ω. As can be seen in Figure 15, under these conditions, the masses move to their equilibrium position close to a separation of $120^{\circ }$ between them. This first case validates that the estimated system behaves as visually (and intuitively) observed when left in rest. It should be noted that this kind of experiment, i.e., a transient, has not been used in the estimation of the parameters but is, however, very well reproduced by the simulations with those parameters.

Two scenarios are evaluated below, but now under stationary excitations similar to those used in the identifications process, with the aim of observing that the extracted power is close to that measured in reality.

In Figure 16, the system is excited with an acceleration of frequency 1 Hz and amplitude 2 m/s^2^ with a load resistance of R=100Ω with the intention of reproducing the case presented in the previous section and shown in Figure 11. In Figure 16, one can also see the time evolution of the azimuthal position of the masses starting from an equilibrium position. It can be clearly seen that the oscillatory movements of the masses (mass 3 can be seen very well) have a frequency similar to that of excitation, as expected. As one can see in Figure 16, the generated power has a shape similar to that expected (see Figure 13). In addition, the maximum power values are very close to those measured in this experiment, and the same is true for the total energy generated in five seconds, close to 27.0 mJ. It should be noted that the extracted power does not have exactly the same shape as in the real case because the initial conditions of the simulation are not the same as in the real experiment; i.e., in both cases, the masses are initially oriented in different azimuthal positions. However, as the estimated generated energy is almost the same, in can be concluded that the estimated model satisfactorily reproduces the measured data.

As noted throughout the paper, the harvester at hand is intended to be excited at low frequencies. Thus, the following simulation is performed with an acceleration of frequency 0.5 Hz and amplitude close to 1 m/s^2^ with the same load resistance of R=100Ω. These values correspond to some of the experiments performed. Figure 17, as in the previous case, shows in the upper part the excitation acceleration, in the middle part the temporal evolution of the masses and, finally, in the lower part the generated power. In this case, we can observe the small displacement to which the masses are subjected, with a maximum range close to $15^{{}^{\circ}}$. Consequently, the extracted power is lower than in the previous case, with maximum peaks close to 1.1 mW. Again, if we compare the power extracted in this case with the same corresponding to one of the experiments, depicted in Figure 14, both share the same shape and the same maximum peak values. In this case, the energy extracted in five seconds is close to 0.7 mJ, as expected, substantially lower than at higher-frequency excitations. In the experiments carried out at 0.5 Hz frequency, the total energy extracted is close to 1.6 mJ. As aforementioned, it should be noted that the extracted power does not have exactly the same shape as in the real case because of the initial conditions of the simulation. Thus, it can be concluded again that the estimated model is able to predict the generated energy in a satisfactory way.

Finally, as the initial idea of the work has been to characterise this system to be used in wind turbines, a simulation with real acceleration signals of a 5 MW wind turbine was performed. These accelerations are shown in the upper part of Figure 18. In this case, the accelerations are bidirectional since so is the motion of the nacelle. This fact enables extracting more energy than if they were uni-directional. However, as can be seen in the figure, due to the wideband nature of the excitation, its amplitude is lower than in the previously shown cases, a fact that contributes negatively to the instantaneous power obtained, even if the frequency is close to 0.5 Hz. The same figure shows the time evolution of the azimuthal position of the masses, as well as the instantaneous power generated. In this case, the total energy extracted during five seconds is close to 0.52 mJ, slightly lower than in the previous case due to the differences in the acceleration signal mentioned above.

### 4.5. Brief Efficiency Comparison with Other Techniques

Although it is not the objective of this paper to show the efficiency of the harvester but rather to describe a procedure for its characterisation, it can be informative to provide basic benchmarking. A more in-depth analysis can be found in [13]. There are very few harvesters reporting experimental data below 1 Hz. Two are based on piezoelectric [38,39] transduction, while another makes use, as with ours, of EM transduction [40]. In this last case, reference is made to the application to wind towers, as in this more recent paper of the same authors [41], although the device is characterised at 2 Hz and not tested under the real or close to real conditions of operation of a WT.

Ref. [40] describes a device providing 60 μW at 0.8 Hz for an acceleration of 0.06 g. Our device provides experimentally 327 μW, a fivefold increase, at similar accelerations even at a lower frequency (0.6 Hz). We have to recognise that our harvester is bulkier. Human movement, and even wind towers, are mentioned as target applications, but the lack of multidirectionality limits the possible use in wind towers to the nacelle. In Ref. [39], 1.6 mW are reported at 0.9 Hz and 0.23 g, versus our 3 μW at exactly the same frequency and lower accelerations, 0.16 g. Their device has much larger dimensions. The proposed device is in this case multidirectional but intended to extract energy from ocean waves. Furthermore, this device is the closest to ours in terms of operation bandwidth. Finally, in Ref. [42], the authors report 3.77 mW average power for a human-induced motion frequency range between 0.9 and 1.93 Hz, and 0.12 g excitation. This is a “broadband” excitation that takes advantage of the higher efficiency at high frequencies. In our case, we can achieve a value of 9 mW with only 0.3 g and a frequency of 1.2 Hz. Our results are, in absolute terms, better, considering that they correspond to a pure harmonic excitation. The device, very imaginative, is based on piezoelectric transduction and can capture energy in every direction. It is, however, bulky and heavy. In a recent paper [43], the authors propose an innovative harvester, also intended to capture energy from human movement, based on triboelectricity. Although it could work at frequencies below 1 Hz, no information is available on the power generated at such frequencies to make a fair comparison.

## 5. Conclusions

We have proposed a comprehensive method to fully characterise a low-frequency, multidirectional vibration harvester based on the electromagnetic principles. First of all, the harvester is modelled taking into account the kinetic energy of the moving masses, the damping introduced by the electromagnetic interaction, the damping due to friction, and the elastic energy introduced by the magnet’s repulsion. This last effect is modelled by either a physical model (white box) or a novel parametric model based on a Fourier series (black box).

An experimental setup is then designed to characterise not only the input (vibration)–output (energy) behaviour but also the internal dynamics of the moving masses and the energy generation mechanisms. The setup includes an image-based acquisition system to measure the masses’ movement and a synchronised signal acquisition system to measure the power generated at each one of the coils.

Then, a set of measurements have been carried out, followed by an identification procedure to estimate the model’s parameters. The identification method extends both to the internal and external (input–output) behaviour of the harvester. The procedure is validated with independent measurements not used for the identification.

Therefore, the method proposed comprises all the steps that should ideally be followed for the complete characterisation of a harvester. Moreover, although we have focused on a particular harvester, the general approach and even many of the experimental and identification methods proposed can be applied to similar harvesters.

Obviously, one of the limitations of our approach is that the modelling of the magnetic interaction, i.e., the mechanical stiffness, depends on the particular geometry of the harvester. In our case, we have proposed two alternatives that adapt to the cylindrical symmetry of the device: one is based on a physical model, and the other is an ad hoc parametric model that renders better results. For other geometries, physical or black box approaches can be introduced in the global model, such as those listed in Ref. [5]. The identification procedure can be applied in the same way, with the only restriction that the model has to be linear for the parameters to be identified no matter the number (complexity). The variables measured will have to be changed to linear displacements (velocities and accelerations) instead of angular displacements as the case may be. Moreover, since the procedure is based on a video acquisition, the accuracy depends on the relative speed of the moving parts. In the case of ultra-low frequencies, this is not a problem.

## Figures and Tables

**Figure 1 sensors-24-03813-f001:**
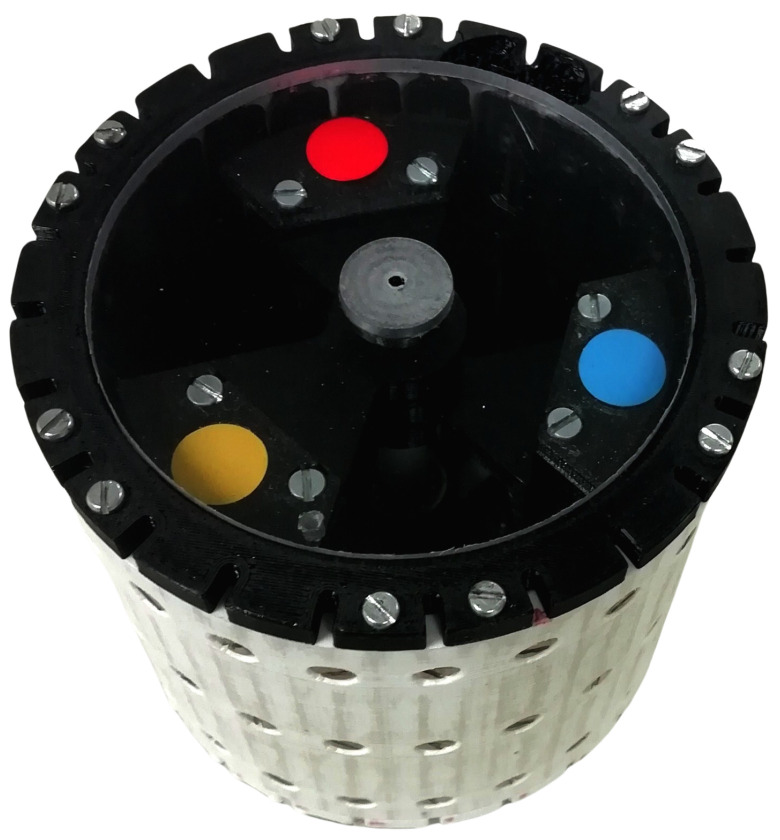
View of the device. Each of the masses is marked with a different coloured sticker.

**Figure 2 sensors-24-03813-f002:**
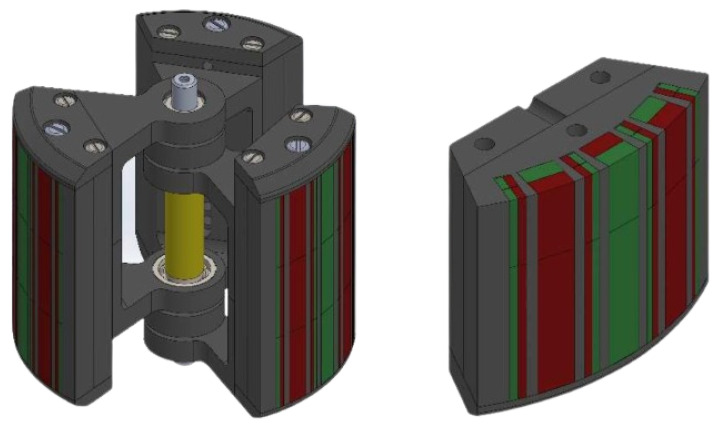
Two perspectives of the Halbach array-equipped moving masses.

**Figure 3 sensors-24-03813-f003:**
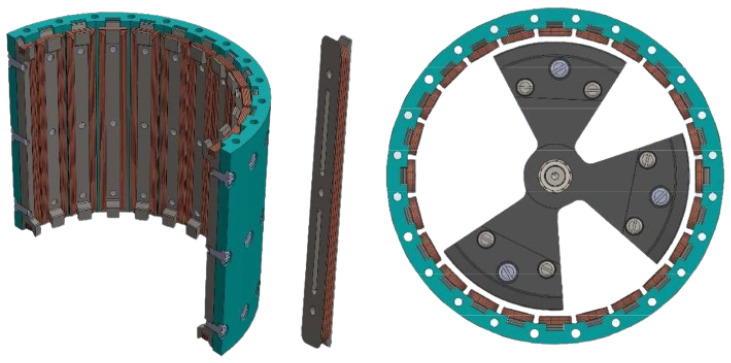
(**Left**): half case with coils. (**Centre**): one coil. (**Right**): upper view with three masses and casing.

**Figure 4 sensors-24-03813-f004:**
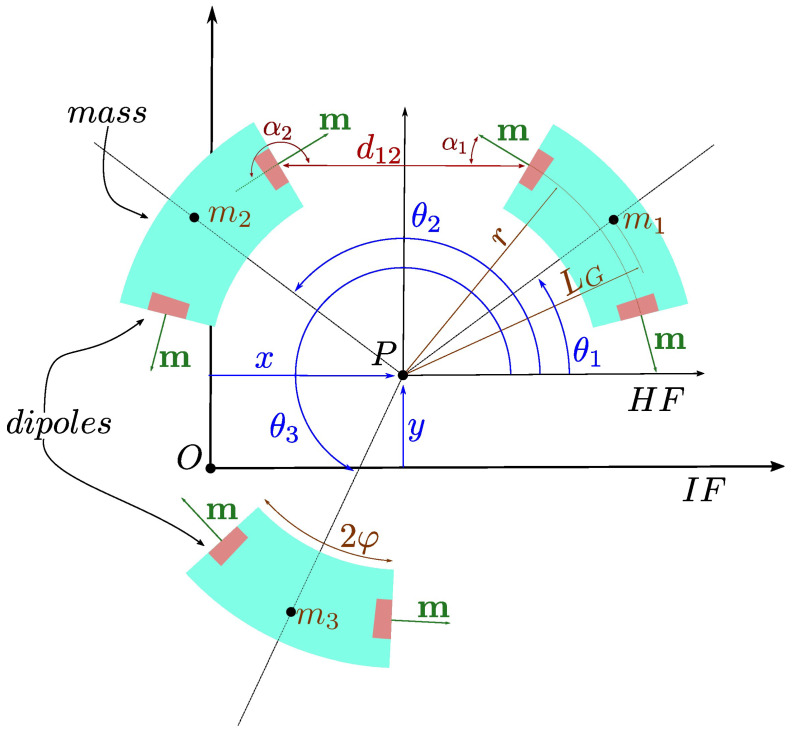
Scheme of the device for kinematics and dynamics. Coordinates *x* and *y* dimension the position of the centre of the device with respect to the IF, and coordinates $\theta_{1}$, $\theta_{2}$ and $\theta_{3}$ dimension the rotation of the masses. Vector m represents the dipoles, and $\alpha_{1}$ and $\alpha_{2}$ are the angles between the segment of length $d_{12}$ and two dipoles of masses 1 and 2, respectively.

**Figure 5 sensors-24-03813-f005:**
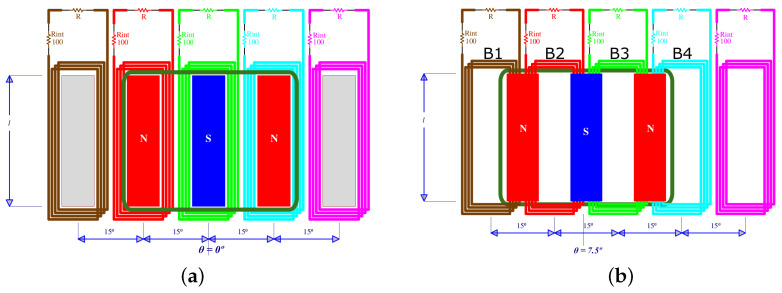
Scheme of the relative positions of magnets and coils for (**a**) minimum and (**b**) maximum power generation.

**Figure 6 sensors-24-03813-f006:**
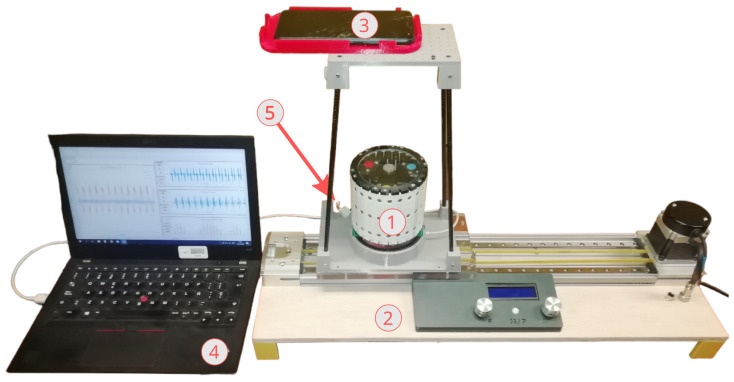
Photo of experimental setup: ➀ energy harvester, ➁ moving platform, ➂ phone with high-speed camera, ➃ computer and ➄ synchronisation LED.

**Figure 7 sensors-24-03813-f007:**
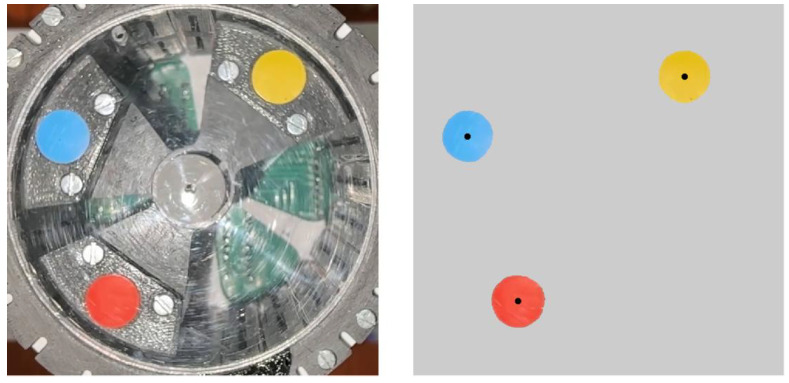
Stickers attached to moving masses and their centre position estimated.

**Figure 8 sensors-24-03813-f008:**
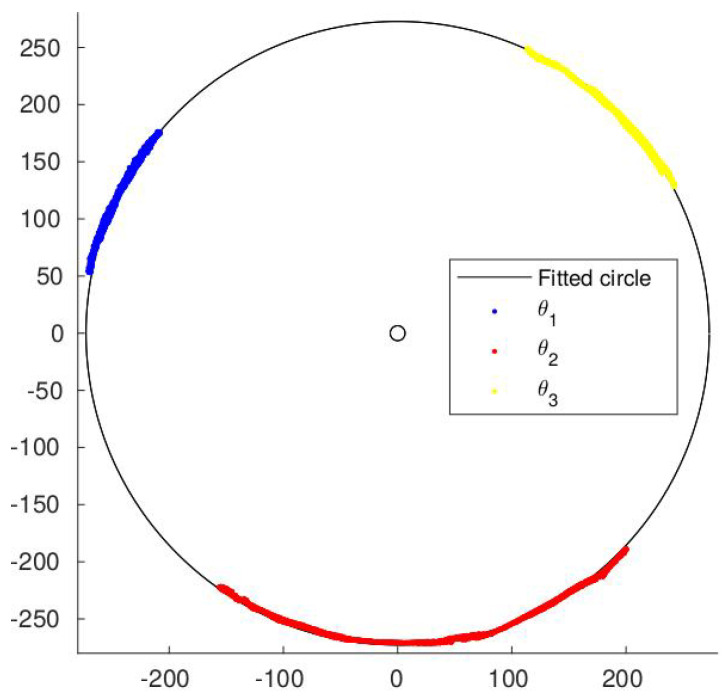
Measured points and the fitted circle. Axes are in pixels and centred at (0,0).

**Figure 9 sensors-24-03813-f009:**
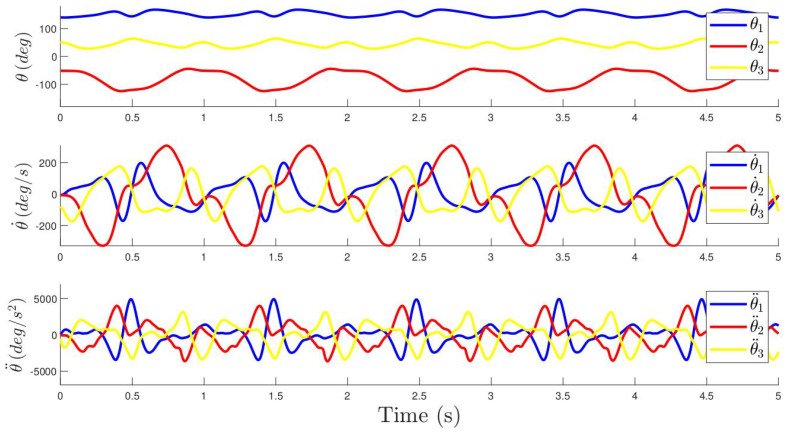
Position, velocity and accelerations of the azimuthal positions of the masses. Each colour indicates the corresponding mass as shown in Figure 7.

**Figure 10 sensors-24-03813-f010:**
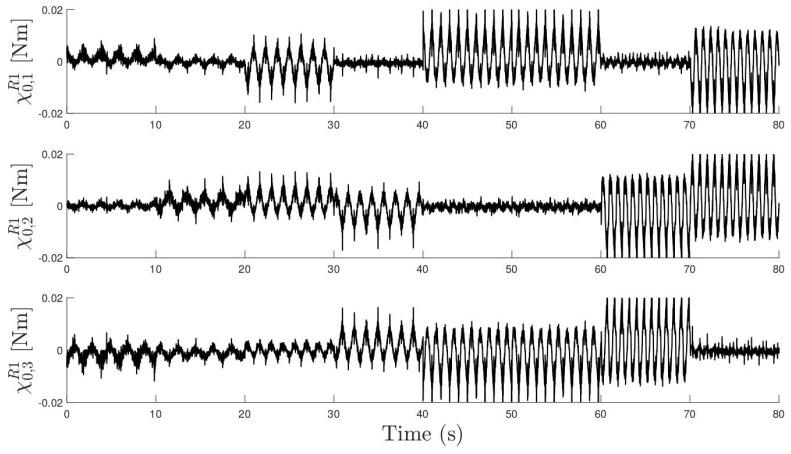
Concatenated time series of $\chi^{est}$ with resistance R=100Ω. Each row represents one of the three components of vector $\chi^{est}$.

**Figure 11 sensors-24-03813-f011:**
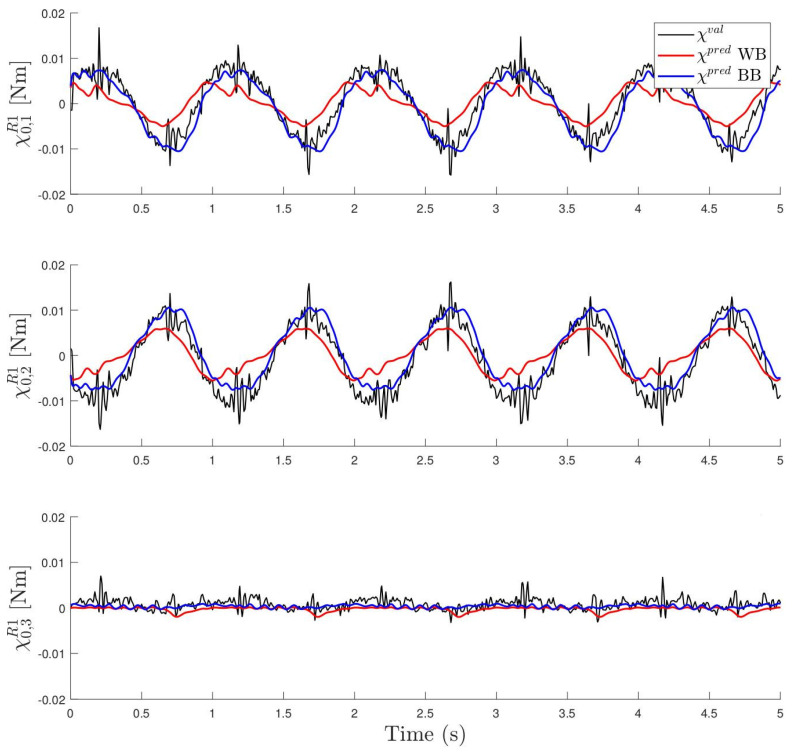
Estimated $\chi_{0}^{R1}$ with the two models (BB and WB) for one case with resistance R=100Ω and frequency 1.0 Hz. In black, measured $\chi^{val}$ is also shown for comparison.

**Figure 12 sensors-24-03813-f012:**
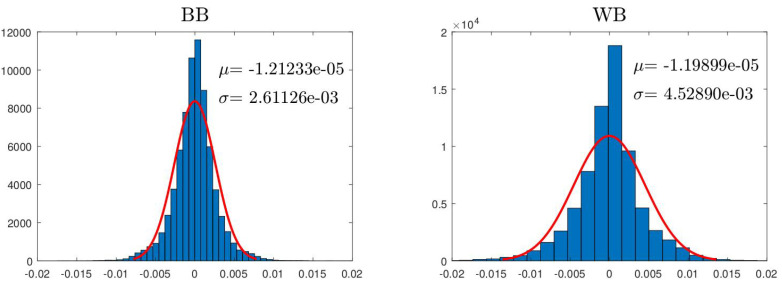
Histograms showing the difference between measured $\chi^{val}$ and the estimated $\chi^{pred}$ for each one of the models.

**Figure 13 sensors-24-03813-f013:**
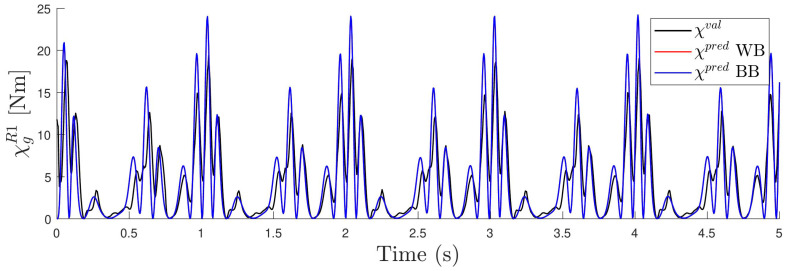
Estimated power $\chi_{g}^{R1}$ with the two models (BB and WB) for one case with resistance R=100Ω and frequency 1.0 Hz. In black, measured power $\chi^{val}$ is also shown for comparison.

**Figure 14 sensors-24-03813-f014:**
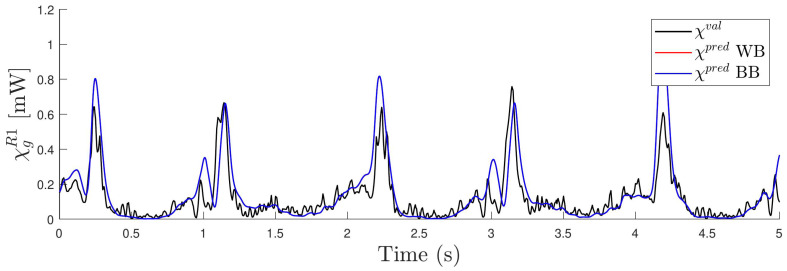
Estimated power $\chi_{g}^{R1}$ with the two models (BB and WB) for one case with resistance R=100Ω and frequency 0.5 Hz. In black, measured power $\chi^{val}$ is also shown for comparison.

**Figure 15 sensors-24-03813-f015:**
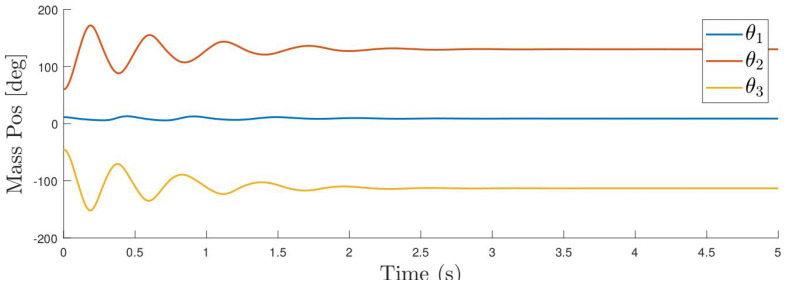
Static simulation performed with the estimated parameters.

**Figure 16 sensors-24-03813-f016:**
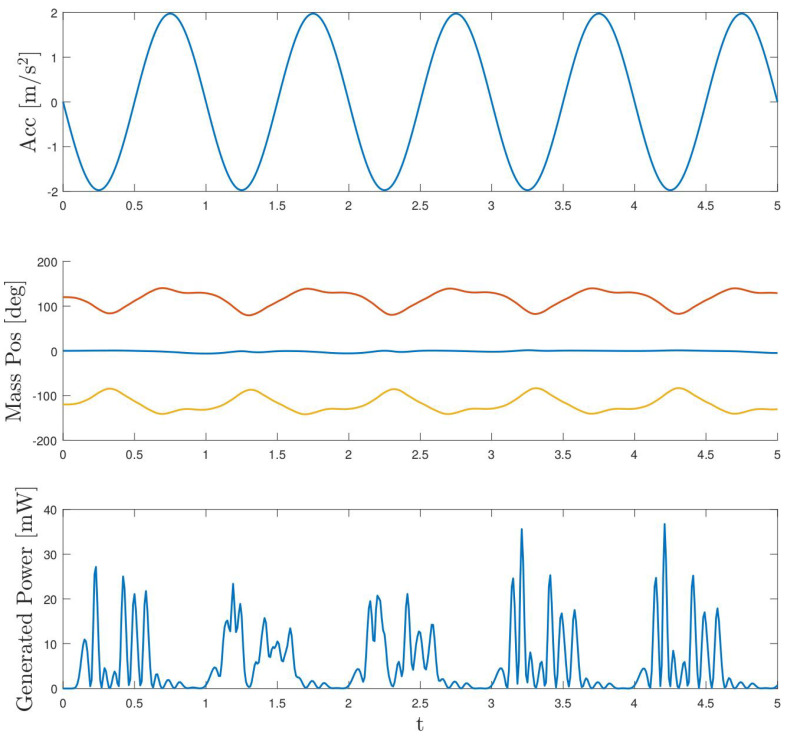
Sinusoidal simulation (1 Hz) performed with the estimated parameters.

**Figure 17 sensors-24-03813-f017:**
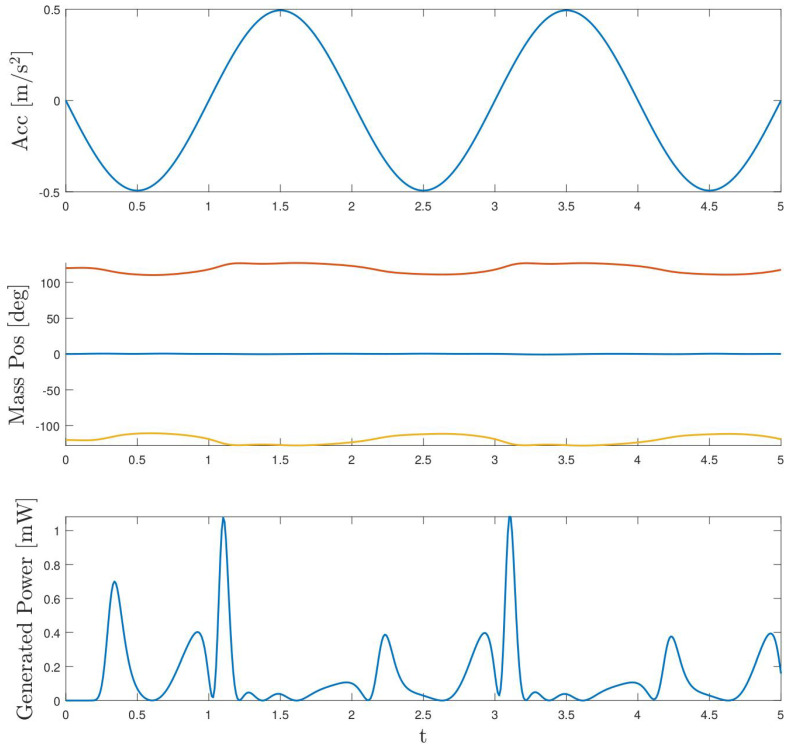
Sinusoidal simulation (0.5 Hz) performed with the estimated parameters.

**Figure 18 sensors-24-03813-f018:**
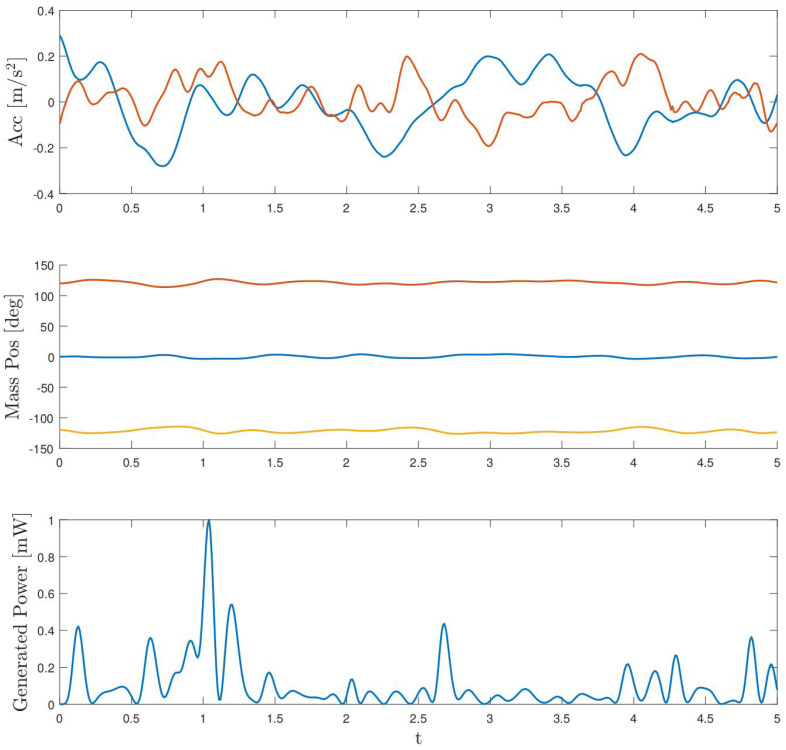
Simulation with wind turbine bidirectional accelerations in blue fore–aft acceleration and in red side-to-side acceleration.

**Table 1 sensors-24-03813-t001:** Numerical values of the estimated parameters.

	BB	WB
$I_{eq}$ [kg m^2^]	1.51 $\times \;10^{4}$	5.42 $\times \;10^{5}$
$b_{0}$ [N m s]	4.31 $\times \;10^{1}$	5.11 $\times \;10^{1}$
$b_{g_{1}}$ [N m s]	7.42 $\times \;10^{1}$	7.42 $\times \;10^{1}$
$b_{g_{2}}$ [N m s]	2.37 $\times \;10^{1}$	2.36 $\times \;10^{1}$
$b_{g_{3}}$ [N m s]	1.81 $\times \;10^{1}$	1.81 $\times \;10^{1}$
c [J, J m^3^]	3.32 $\times \;10^{3}$, −2.75 $\times \;10^{2}$, 4.58 $\times \;10^{3}$, −3.17 $\times \;10^{3}$	7.75 $\times \;10^{5}$
	6.22 $\times \;10^{3}$, −3.06 $\times \;10^{2}$, −1.07 $\times \;10^{3}$, −5.01 $\times \;10^{3}$	
	−2.05 $\times \;10^{3}$, −6.11 $\times \;10^{3}$, −1.14 $\times \;10^{3}$, −5.97 $\times \;10^{3}$	

## Data Availability

The raw data supporting the conclusions of this article will be made available by the authors on request.

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
