# Peer review of "Comprehensive Characterisation of a Low-Frequency-Vibration Energy Harvester"

_sensors, 2024, doi:10.3390/s24123813_

Round 1

Reviewer 1 Report

Comments and Suggestions for Authors

This manuscript reports the detailed characterazation of a low-frequency energy harvester reported in ref. [13]. I have some suggestions below:

1. Please provide the efficiency comparison with other techniques, such as piezo-related devices.

2. Please compare the model and test setup herein with that in ref. [13]. They look similar to me.

Reviewer 2 Report

Comments and Suggestions for Authors

The manuscript conducted dynamic and experimental research on electromagnetic vibration energy harvesters, and the proposed model has certain research significance. Rich research work, fully revealing the performance of energy harvesters, with strong innovation. However, the research on relevant theories and experiments has not been mutually validated, so it is recommended to make modifications to this. Regarding research on energy harvesters, it is recommended that the author pay attention to the following literatures: Mechanical Systems and Signal Processing, 2024, 212, 111304.

Reviewer 3 Report

Comments and Suggestions for Authors

The author describe a measurement procedure to fully characterize a novel vibration energy harvester, operating in the ultra-low frequency range in this paper.

1. In page18 “5. Conclusion”, the author said:”We have proposed a comprehensive method to fully characterize a low-frequency, multidirectional, vibration harvester based on the electromagnetic principle.”, at refer to the experiment setup in Figure 4(page 9), I can not see any evident how to test the multidirectional function of the proposed EM harvester.

2. Since 24 coils are used in the proposed harvester, so how to collect all the output power of the 24 coils generator Simultaneously. Because the current in the coils may differed in the same, how to make sure the whole output power to make stable?

3. In page 8”a triaxial accelerometer (ADXL345) is installed in the electronics of the harvester”, but I can not see the measurement data for the accelerometer, can you explain it and tell me the function of the accelerometer?

4. Since some equations and content are similar to your publish article”Ultra-low frequency multidirectional harvester for wind turbines””https://doi.org/10.1016/j.apenergy.2023.120715 ”, you should give some cite to your published article in this article.

Reviewer 4 Report

Comments and Suggestions for Authors

This article gives an interesting development in energy harvester, However, there are still some comments in the following to be addressed.

1. As given in conclusions, the authors should give a diagram on how to convert mechanical energy to and electromagetic energy, or something else.

2. In (23), how accurately to detect the signals of X (torque?)? This affacts the accuracy of the estimation.

3. Possible applications by the developed energy harvester should be described.

Reviewer 5 Report

Comments and Suggestions for Authors

This paper details a method to characterize a low-frequency, multidirectional electromagnetic vibration harvester. The harvester is modeled considering kinetic energy, damping, and elastic forces using both physical (White Box) and parametric (Black Box) approaches. An experimental setup measures mass movements and power generation.

1.     Did the authors compare the magnetic potential energy with other methods?

2.     In Fig. 9, the legend for the last subfigure is incorrect.

3.     For Figs. 9-16, indicating which figures represent experimental data and which are simulations would enhance readability.

4.     On page 12, the authors state, "Moreover, these experiments are performed at different excitation frequencies (500, 750, 1000, and 1200 mHz) to validate the model across the device's designed frequency range. Additionally, these experiments are repeated for several electrical resistances (100Ω, 666Ω, and 1kΩ) to observe varying power generation coefficients." However, not all experimental results are presented.

5.     Rather than using a fixed excitation frequency, would it be feasible to conduct frequency response tests, such as frequency sweeps, to provide a more comprehensive analysis?

Overall, this reviewer recommends a major revision before considering this manuscript for publication.

Round 2

Reviewer 4 Report

Comments and Suggestions for Authors

I know the power can be generated under the Faraday's Law. However, from Figs. 1 and 4, it is difficult to know how to generate the electric power by vibrating. Maybe a mechanical structure diagram about how to convert the vibrating energy to electric power energy should be given.

Author Response

Dear reviewer,

It is true that we had only detailed the process of energy extraction in the text. We have added a couple of pictures to the article (the new figures 2 and 3) that complement the text to explain the process of transforming vibration energy into electrical energy. The text has been updated (in green) with reference to those images. 

Best regards,

The authors

Reviewer 5 Report

Comments and Suggestions for Authors

i dont have any further rejection for the paper with respect to the vibration and energy harvesting part. revise

Author Response

Dear reviewer,

We are pleased to have responded to all the questions.